# Involvement of the Gut Microbiome in the Local and Systemic Immune Response to Pancreatic Ductal Adenocarcinoma

**DOI:** 10.3390/cancers16050996

**Published:** 2024-02-29

**Authors:** James M. Halle-Smith, Hayden Pearce, Samantha Nicol, Lewis A. Hall, Sarah F. Powell-Brett, Andrew D. Beggs, Tariq Iqbal, Paul Moss, Keith J. Roberts

**Affiliations:** 1Hepatobiliary and Pancreatic Surgery Unit, Queen Elizabeth Hospital Birmingham, Birmingham B15 2GW, UK; 2Institute of Immunology and Immunotherapy, University of Birmingham, Birmingham B15 2TT, UK; 3Institute of Cancer and Genomic Sciences, University of Birmingham, Birmingham B15 2TT, UK; 4Department of Gastroenterology, Queen Elizabeth Hospital Birmingham, Birmingham B15 2GW, UK; 5Microbiome Treatment Centre, University of Birmingham, Birmingham B15 2TT, UK; 6National Institute for Health Research Birmingham Biomedical Research Centre, Birmingham B15 2TT, UK

**Keywords:** pancreatic ductal adenocarcinoma, immunosuppression, gut microbiome, tumour microbiome

## Abstract

**Simple Summary:**

One of the reasons that pancreatic cancer is such a deadly disease is that it is able to evade the body’s usual defence system (the immune system). It has been shown that the population of microorganisms in the human gut (the gut microbiome) plays a role in regulating the human immune system. This article outlines the ways in which the gut microbiome influences the immune system in pancreatic cancer both within the bloodstream (systemic) and around the pancreatic tumour itself (local). These are important mechanisms because greater understanding of these will direct the development of future treatments for pancreatic cancer. It is possible that some of these treatment options will target the gut microbiome in order to boost the immune system’s response to pancreatic cancer.

**Abstract:**

The systemic and local immunosuppression exhibited by pancreatic ductal adenocarcinoma (PDAC) contributes significantly to its aggressive nature. There is a need for a greater understanding of the mechanisms behind this profound immune evasion, which makes it one of the most challenging malignancies to treat and thus one of the leading causes of cancer death worldwide. The gut microbiome is now thought to be the largest immune organ in the body and has been shown to play an important role in multiple immune-mediated diseases. By summarizing the current literature, this review examines the mechanisms by which the gut microbiome may modulate the immune response to PDAC. Evidence suggests that the gut microbiome can alter immune cell populations both in the peripheral blood and within the tumour itself in PDAC patients. In addition, evidence suggests that the gut microbiome influences the composition of the PDAC tumour microbiome, which exerts a local effect on PDAC tumour immune infiltration. Put together, this promotes the gut microbiome as a promising route for future therapies to improve immune responses in PDAC patients.

## 1. Introduction

The gut is now considered the largest immune organ in the human body [1,2,3,4,5]. This likely derives from the intense relationship between immune cell populations and the gut microbiome: the diverse population of microorganisms (microbiota) and their associated genetic material resident within the gastrointestinal tract [1,2,3]. Widespread availability of bacterial genome sequencing, such as 16s rRNA and metagenomic sequencing [6,7,8,9], has allowed researchers to begin to unravel the complex pathways in which these microorganisms, including bacteria, viruses and archaea, influence metabolic and immune functions of their human host [1,2,3,4,5].

The gut microbiome has been implicated in the modulation of multiple immune-mediated disease processes, showcasing its influence across diverse conditions. A notable example is inflammatory bowel disease (IBD) [10,11,12] where fluctuation of gut bacteria and their metabolites is related to clinical state in both ulcerative colitis and Crohn’s disease [13]. Furthermore, the immunomodulatory nature of the microbiome extends beyond gastrointestinal disorders, impacting chronic liver disease [14,15], graft versus host disease (GvHD) [16], lupus nephritis [17,18] and various malignancies [19,20]. Inflammation plays an important role in cancer development and as such it is not unexpected that the gut microbiome may be a critical determinant, potentially most notably for tumours of the gastrointestinal tract. Indeed, it is now becoming clear that the gut microbiome may play a role in the development and progression of pancreatic ductal adenocarcinoma (PDAC). Indeed, gut microbiome diversity has been shown to be different in short-term survivors (STSs) and long-term survivors (LTSs) of PDAC, with greater diversity linked to improved outcomes [21]. This influence is thought to occur through its impact on the local and systemic immune system [22,23], and makes the gut microbiome a promising avenue for future investigations. There is a need for a greater understanding of the mechanisms behind the profound immune evasion exhibited by PDAC tumours [24], one of the many contributing factors that make it one of the most challenging malignancies to treat, and thus one of the leading causes of cancer death worldwide [25,26,27]. The gut microbiome may play a leading role in this regard and is the subject of this review, which focuses on the mechanisms through which the microbiome is thought to modulate the immune response to PDAC.

## 2. The Integral Relationship between the Gut Microbiome and the Immune System

The development of a healthy immune system is dependent upon the trillions of organisms that inhabit the human gut, collectively known as the gut microbiota [28,29,30,31,32]. This symbiotic relationship is highly dynamic, as seen in studies involving animals with compromised immune systems, where the reintroduction of commensal bacteria leads to a resurgence of immune capabilities [29,30]. However, studying the interplay between the gut microbiota and the human immune system is more challenging. Valuable insights have emerged from the study of patients recovering from haematopoietic stem cell transplantation. For example, the immune cell dynamics of patients recovering from chemotherapy and stem-cell engraftment was reviewed in association with their gut microbiome [33]. Importantly, those who received auto-faecal microbiota transplantation (FMT) displayed accelerated immune system reconstitution compared to control groups, indicating a strong correlation between a healthier gut microbiome and swifter immune recovery [33]. Further evidence of a close relationship between the gut microbiome and the systemic immune system comes from studies of a murine bone marrow transplantation (BMT) model. Here, reduced diversity within the gut microbiome negatively impacted the repopulation of neutrophils and lymphocytes, but not cells of the erythroid lineage [34].

Bacteria in the gut interact with the host immune system through a number of different mechanisms, with a phylogenetically diverse gut microbiota being required for the development of a robust immune system [35]. One example of this is through presentation of bacterial polysaccharide (PSA) by dendritic cells in the intestine, which, in turn, activates CD4+ T cells and subsequent cytokine production [36]. Importantly, also, this pathway is implicated in maintaining the delicate balance between Th1/Th2 T cells, essential for optimal immune function [36]. A further example of immune homeostasis in the gut is that of Th17 cells. It has been shown that Th17 cell differentiation in the bowel mucosa is dependent upon specific commensal bacteria [30,37]. Their presence in the bowel mucosa occurs only following colonization by particular commensal bacteria, while their differentiation is hindered by antibiotic treatment [30]. One study reported that the differentiation of Th17 cells within bowel mucosa was correlated to the presence of cytophaga–flavobacter–bacteroides (CFBs) [30]. Additionally, microbiota-derived metabolites, such as the short-chain fatty acid (SCFA) butyrate, have been shown to influence immune cell populations in the GI tract with broader systemic implications [16,38,39].

## 3. The Impact of the Gut Microbiome on the Systemic Immune Response in PDAC

The systemic immunosuppression exhibited by PDAC contributes significantly to its aggressive nature [40,41,42,43,44,45]. Recent studies have shown that systemic adaptive immunity plays an important role in mediating the relationship between the gut microbiome and progression of PDAC. In one murine PDAC study, depletion of the gut microbiome through antibiotic administration resulted in notably slower pancreatic tumour growth and progression compared to control groups with an intact gut microbiome [22], similar to findings of other studies [23]. Interestingly, this tumour-suppressing effect was not observed in Rag1 knockout mice lacking mature T and B lymphocytes, suggesting the role of systemic adaptive immunity in this interaction [22].

Moreover, investigations involving IL-17, the primary cytokine released by Th17 cells, revealed crucial insights. Mice treated with an IL-17 neutralizing antibody did not exhibit the same reduction in pancreatic tumour growth following gut microbiome depletion [22]. This highlights an important role for IL-17 in the interaction between the gut microbiome and pancreatic tumour progression [22]. Th17 cells are pivotal in inflammatory responses and mucosal defence against pathogens [46,47,48], and rely on specific commensal bacteria for their differentiation in the bowel mucosa [30]. Sequencing of faecal samples from mice in which the gut microbiome was depleted using antibiotics, leading to a reduction in pancreatic tumour growth, showed a reduction in the abundance of the *Bacteroidetes* phylum; the same taxa implicated in the differentiation of Th17 cells in the gut [22,30]. Interestingly, a study that reported a specific faecal microbiome signature for PDAC revealed that a reduction in the species *Bacteroides caprocola* was a sensitive and specific change in the faecal microbiome of PDAC patients [49]. Consequently, it is possible that a reduced abundance of *Bacteroides* acts to limit Th17 differentiation and IL-17 production. Indeed, peripheral blood analyses in pancreatic cancer patients reveal diminished concentrations of Th17 cells and IL-17 compared to healthy counterparts [50], with further declines in more advanced stages of the disease [50].

## 4. The Role of the Gut Microbiome in Modulating Immune Infiltration in PDAC Tumours

One of the contributing factors to the poor prognosis of PDAC is immune evasion, which has been observed in the tumour microenvironment [40,51]. As such, the composition of immune cells in the tumour microenvironment has been shown to be a prognostic factor in pancreatic cancer [52]. Depletion of the gut microbiome with oral antibiotics leads to reduced growth of PDAC tumours in murine models [22,23] and this associates with increased T-cell infiltration and a decrease in myeloid-derived suppressor cells (MDSCs) (Figure 1). This is of clinical relevance, since lower T-cell infiltration is associated with a poorer prognosis [53]. Specifically, an increased intra-tumoural CD8:CD4 T cell ratio has been associated with increasing immunogenicity of PDAC [54] and better survival in PDAC patients [52], likely due to higher neoantigen load. Furthermore, the presence of MDSCs within PDAC tumours is inversely proportional to tumour-infiltrating effector T cells [55], so a reduction in their level after ablation of the microbiome is also of clinical relevance. An increased intra-tumoural Th1:Th2 cell ratio has also been observed following microbial depletion [22]. Importantly, this correlated with improved survival [56]. A reduction in Th17 cells has also been seen in the microenvironment of pancreatic tumours in mice who had their gut microbiome depleted with antibiotics [22]. IL-17, the primary cytokine released by TH17 cells, has been associated with initiation and progression of pancreatic intraepithelial neoplasia (PanIN) [57], so a reduction in the proportion of Th17 cells in the TME is desirable [22]. Indeed, studies of human PDAC patients have shown substantial expansion of Th17 cells in the tumour [51]. This is relevant given that an increased proportion of Th17 cells in tumours is associated with poorer clinical outcome [58] (Figure 1).

Collectively, these findings suggest that the one of the mechanisms by which the gut microbiome influences pancreatic tumour progression is through the modulation of tumour immune infiltration (Figure 1). Further evidence for a dynamic role of the gut microbiome in PDAC progression came from further experiments in which the gut microbiome of murine models was changed using faecal transplant via oral gavage. When faeces from wild-type (WT) mice was used, there was no acceleration in tumourigenesis and T-cell infiltration remained unaffected. However, repopulating the gut microbiome with faeces from genetically predisposed PDAC mice [*Pdx1*^Cre^; *LSL-Kras*^G12D^; *Trp53*^R172H^ (KPC)] accelerated tumour growth and decreased T-cell infiltration [23]. This implies that genetically susceptible mice harbour gut bacteria that are capable of increasing oncogenesis through modulation of T-cell infiltration.

A further mechanism by which bacteria of the gut microbiome may influence the systemic immune response to PDAC is through their metabolites. One such example is short-chain fatty acids (SCFAs), such as butyrate, which are able to exert an anti-inflammatory response in distant tissues [59,60,61]. In addition, the lactic acid produced by some bacteria of the gut microbiome, such as *Lactobacillus plantarum* 299, have been shown to play a role in the regression of carcinogenesis through the modulation of the immune system [61,62].

## 5. The Influence of the Gut Microbiome on Tumour Microbiome Composition

A route by which the gut microbiome is thought to influence the tumour immune infiltration is through translocation of bacteria from the gut to the pancreas. Historically, tumour tissue was believed to be sterile; however, it is now recognized that various cancer types harbour a bacterial population termed the tumour microbiome [63,64]. Breast and pancreatic tumours have been shown to have a particularly diverse microbiome [64]. Notably, the composition of the PDAC tumour microbiome has been associated with prognosis [21,23], which is similar to other gastrointestinal tumour types such as cholangiocarcinoma [65], neuroendocrine [66] and colorectal cancers [67,68]. In pancreatic cancer, distinct microbial communities have been identified within tumour areas compared to adjacent pancreatic tissues [21,23,64,69]. Moreover, it has been shown that the gut microbiome influences the composition of this tumour microbiome [21,23].

This interaction is hypothesised to occur via several routes, including the portal circulation, mesenteric lymph nodes, or directly through the biliary and pancreatic duct systems [69,70,71]. One study investigated this directly in a murine model by introducing fluorescently labelled bacteria into the gut microbiome via oral administration. Subsequent fluorescence in situ hybridization (FISH) confirmed the presence of these labelled bacteria within pancreatic tumours, indicating their translocation from the gut [23]. Crucially, emerging evidence suggests that these bacterial populations within tumours might influence the immune infiltration of PDAC tumours.

## 6. Impact of the Tumour Microbiome on PDAC Tumour Immune Infiltration

As highlighted earlier, depletion of the microbiome in murine models led to improved immune infiltration within PDAC tumours. Notably, T cells emerge as pivotal players in this phenomenon as this effect is not seen when T cells are depleted [23]. Transcriptomic analyses also confirmed that genes associated with T-cell proliferation and immune activation were upregulated in PDAC tumours following antibiotic ablation [23]. The hypothesis that T cells mediate the reduction in tumour growth following microbiome ablation with antibiotics has been further tested by transferring intra-tumoural T cells to murine subjects challenged by subcutaneous pancreatic tumour [23]. When T cells from mice that had been treated with antibiotics were introduced to the subcutaneous pancreatic tumour, a reduction in tumour burden was observed [23]. Contrastingly, when the T cells from tumours from mice that had not been treated with antibiotics were transferred, no difference in tumour growth was observed [23]. These findings imply that microbiome depletion may trigger the activation of intra-tumoral T cells, which contributes to reducing tumour growth.

### 6.1. Impact of the Tumour Microbiome on T-Cell Activation within the Tumour Microenvironment

Investigation into the mechanism by which the microbiome mediates T-cell infiltration in PDAC implicates a role for macrophages. Macrophages act as central regulators of T-cell activation, for example by presenting foreign antigen via major histocompatibility complex (MHC) class II to T cells [72,73,74]. However, for effective T-cell activation, additional complex costimulatory signals are crucial [72,73,74] and, as such, different levels of activity are observed in distinct environments. Notably, in a study where macrophages were exposed to bacterial extracts from the gut microbiome of PDAC mice and control mice, macrophages exposed to PDAC gut microbiome demonstrated reduced CD4+ and CD8+ T-cell activation, alongside impeding Th1 differentiation of CD4+ T cells. In contrast, macrophages exposed to gut bacteria from control mice did not exhibit the same inhibitory properties [23]. This suggests that macrophages within PDAC tumours acquire an immunosuppressive role, stimulated by the microbiome. Additionally, tumour-associated macrophages (TAMs) were harvested from the mice that had their microbiome ablated with antibiotics and these showed an increased capacity to activate T cells compared to those from control mice [23]. Further substantiating the active involvement of TAMs in mediating the relationship between the microbiome and immune infiltration, in vivo experiments that neutralized macrophages nullified the observed increased intra-tumoral T-cell activation after antibiotic-induced microbiome ablation [23]. These findings collectively indicate that TAMs actively modulate the interaction between the microbiome and the immune infiltrate within PDAC tumours.

### 6.2. Tumour Microbiome Interaction with Pattern Recognition Receptors (PRRs)

Mechanistic studies show that gut bacteria can potentially contribute to oncogenesis by inducing immunosuppression in the tumour microenvironment through modulation of pattern recognition receptors (PRRs); one of the ways in which bacteria of the microbiome interact with the host immune system [75,76]. Recent investigations have uncovered a role for PRRs in microenvironmental immune regulation in multiple different cancer types [75]. Indeed, it has been shown that if certain PRRs are upregulated in PDAC, this leads to accelerated oncogenesis through immune suppression at the tumour site [77,78,79]. For example, ligation of TLR9 on pancreatic stellate cells [80] and activation of Dectin-1 on macrophages promotes pancreatic tumourigenesis through induction of immune tolerance [81].

A relationship between the microbiome and PRR expression within PDAC tumours has also been demonstrated in some murine models; when the microbiome was depleted, markedly lower expression of PRRs compared to controls was observed [23]. Equally, repopulating the microbiome with faeces from PDAC mice (which led to increased tumour growth in the same study) with concurrent blockade of TLR signalling negated the previously seen increased oncogenesis. The same study group then entrained macrophages with the PDAC microbiome, which previously led to T-cell immunosuppression, but inhibited TLR signalling. These macrophages were then unable to inhibit T-cell activity [23]. Collectively, these findings suggest that the PDAC microbiome can exert tumour-promoting effects by influencing TAMs via TLR signalling, inducing immune tolerance.

### 6.3. Interaction between the Tumour Microbiome and Tumour-Associated Macrophages

It has also been shown that TLR activation can expand MDSC and anti-inflammatory TAM populations within tumours [82]. TAMs have many different roles in the tumour microenvironment [83] and can influence T cell immunogenicity in PDAC [84,85]. Relevant to this review, ablation of the microbiome in murine PDAC models led to differences in tumour-associated macrophages (TAMs), with a reduction in the immune-suppressive M2-like TAM and an increase in the M1-like TAMs [23]. It is thought that M1-like TAMs are associated with better prognosis seen as they express higher MHC II, CD86, TNFα, IL12 and IL6 [84,85]. There was also evidence that those TAMs in PDAC tumours where the microbiome had been ablated expressed increased chemokine levels, vital for appropriate T-cell activation [23,86].

## 7. The Gut Microbiome Influences Response to Systemic Anti-Cancer Therapies

### 7.1. Immunotherapy

While other tumour types, such as melanoma, have shown promising results to immunotherapy, results for its use in PDAC have been disappointing [87,88]. These therapies block immune inhibitory pathways such as CTLA-4 and PD-1 [89,90] and response rates are partially determined by the pattern of endogenous T-cell infiltrate in the tumour microenvironment before therapy [91,92,93]. As detailed above, the gut microbiome modulates the T cell infiltration of PDAC tumours and may thus influence the efficacy of immunotherapy. For example, the gut microbiome plays an important role in the differentiation of Th17 cells and it has been suggested that Th17 cells facilitate resistance to immunotherapies such as anti-PD1 checkpoint blockade [94]. As such, greater understanding of these mechanisms presents opportunities for future therapies.

In human melanoma patients, the gut microbiome of patients that responded to anti-PD-1 immunotherapy has been compared to those who did not respond [95,96]. This showed that the composition of the gut microbiome was significantly different between the groups; with greater alpha diversity in the survivors [95]. In terms of specific species, a greater abundance of *Bifidobacterium longum*, *Collinsella aerofaciens*, *Enterococcus faecium* and *Ruminococcaceae* has been reported amongst responding patients [95,96]. The effect of the gut microbiome on anti-PD-1 efficacy has also been shown to be dynamic in murine studies. Here, authors administered FMT, from cancer patients who had responded to ICI therapy and those that did not, to murine cancer models. They observed mildly increased efficacy of anti-PD-1 in the murine models after FMT of responder faeces but not after FMT of non-responder faeces [97]. Of further interest, a direct correlation between the relative abundance of certain bacteria types, *Akkermansia muciniphila* [97] and *Bifidobacterium* [98], have been observed. The efficacy of CTLA-4 blockade therapy has also been shown to be dependent of the gut microbiome, specifically on the presence of Bacteroides species, such as *B. fragilis* [99]. In this study, they were also able to increase the efficacy of ICI through FMT in murine melanoma models [99].

Returning to PDAC, anti PD-1 therapy had no effect on tumour growth in PDAC control mice [23]. However, when antibiotics were given alongside anti-PD-1 therapy, tumour growth reduced [23]. Excitingly, there was evidence that antibiotic and anti-PD-1 therapy acted synergistically, evidenced by upregulation of CXCR3 and LFA1, which was not seen in the monotherapy groups [23].

### 7.2. Chemotherapy

With the use of next-generation sequencing and metabolomic analysis, differences in the gut microbiome between patients that respond to chemotherapy and those that do not have been identified [100]. Specifically, one group identified that a microbiome derived metabolite, indole-3-acetic acid (3-IAA) was enriched in metastatic PDAC patients that responded to chemotherapy compared to those who did not [100]. Other groups have also hypothesized a link between the gut microbiome and the efficacy of chemotherapy, after observing that antibiotic administration was associated with an improved response to chemotherapy [101,102]. It is thought that altering the composition of the gut microbiome, for example through diet or antibiotics, influences the composition of the tumour microbiome through translocation of bacteria from the intestine to the tumour [21,103]. A further interesting observation is that antibiotic administration improved response to gemcitabine rather than fluorouracil [101,102] and resident bacteria in PDAC tumours have been shown to be capable of metabolizing gemcitabine to its inactive form [69].

## 8. Discussion

These findings suggest that targeting the gut microbiome may provide an opportunity to enhance the efficacy of systemic chemo- and immunotherapy in PDAC. However, it should be acknowledged that the majority of the studies reported in this review are animal or in vitro studies. To move these findings closer to clinical practice, further human studies are required so that attention can turn to how the gut microbiome can safely be modulated in PDAC patients in order to enhance the immune response and thus treatment response.

In order to achieve this the composition of a ‘responder’ gut microbiome phenotype needs to be understood in future studies. Although gut microbial signatures of PDAC in human patients have recently been reported [49], associations between PDAC treatment response and the composition of the gut microbiome have largely been limited to animal studies. There are multiple methods by which the progress of PDAC can be monitored in patients, such as carbohydrate antigen 19-9 (Ca19-9) levels, radiological tumour response or pathological tumour regression grading [104,105]. As such, future prospective studies should seek to correlate the response to systemic treatment with the composition of the gut microbiome.

As detailed above, the gut microbiome has been shown to be related to prognosis and treatment response in PDAC, so efforts should turn to how the gut microbiome can be safely modified towards a more favourable phenotype [106,107] (Figure 2). In some animal studies, oral supplementation with *A. muciniphila* [97] and *Bifidobacterium* [98] improved response to immunotherapy. Encouragingly, technology now exists that allows specific beneficial bacteria strains to be cultured and administered using so-called next-generation probiotics [108]. As such, it may be that in future studies probiotics administered alongside systemic therapy may improve treatment response in PDAC patients. Another possible form of supplementation is through microbiome-derived metabolites. One study showed that administration of trimethylamine N-oxide (TMAO) enhanced anti-tumour immunity to PDAC [109].

FMT has been shown to lead to widespread changes in recipient microbiome, including the establishment of new species [110,111], and has attracted increasing interest due to its efficacy in treating refractory *C. difficile* infection [101]. In addition, murine studies have shown that FMT can influence the immune infiltration of PDAC tumours [21,23,95]. As such, concomitant administration of FMT may potentiate the effect of systemic therapies in PDAC and so may be an avenue to explore in future clinical trials.

Antibiotics can influence the immune infiltration of PDAC tumours in murine studies [22,23]. Clinical studies have also shown that receipt of antibiotics is linked to chemotherapy response and survival in PDAC [102,112]. To investigate this further, the gut microbiome and treatment response of patients who have received antibiotics should be compared to those who have not in future human studies. This will help to determine the changes in the gut microbiome that may mediate these changes in treatment response and survival. Further to this point, it should also be acknowledged that gaining an accurate representation of the gut microbiome in PDAC patients can be difficult due to steps along the diagnostic pathway. For example, jaundice, antibiotics and interventional procedures (such as endoscopic or percutaneous biliary drainage) have the potential to alter the gut microbiome substantially, so future studies should ensure that these factors are controlled for.

A common feature of PDAC is pancreatic exocrine insufficiency (PEI), which is prevalent and progressive, particularly in those patients with tumours located in the head of the pancreas [113,114]. PEI is associated with poorer survival in PDAC and its treatment with pancreatic enzyme replacement therapy (PERT) improves survival almost as much as chemotherapy [115]. It was initially thought that this effect was solely due to alleviating the nutritional depletion and micronutrient deficiencies associated with PEI, allowing patients to withstand more curative treatment [115]. However, it is increasingly understood that PEI can cause dysbiosis in the gut microbiome [115,116]. Importantly, in animal models PERT has been shown to reverse changes to the microbiome caused by PEI [117,118]. Specifically, an increase in beneficial bacterial species has been observed, such as *Akkermansia muciniphila,* which has been associated with better response to immunotherapy [97]. The impact of PEI on PDAC treatment resistance should therefore be better understood through further evaluation of the gut microbiome in human studies.

The main body of evidence supporting a link between the gut microbiome and the immune response to PDAC has come from animal studies. Therefore, to further investigate this relationship in humans, future studies could simultaneously measure the gut microbiome composition and the peripheral and tumour immune response in PDAC patients. This could be achieved by quantifying peripheral immune markers in blood, as well as tumour-infiltrating lymphocytes, alongside a detailed analysis of the gut microbiome.

## 9. Conclusions

The poor survival and treatment resistance of PDAC is characterised by local and systemic immune suppression and the gut microbiome has been shown to influence the immune response to PDAC within both compartments. Knowledge of the pathways associated with this, and the organisms involved, presents a range of opportunities for the development of novel therapies to this challenging condition.

## Figures and Tables

**Figure 1 cancers-16-00996-f001:**
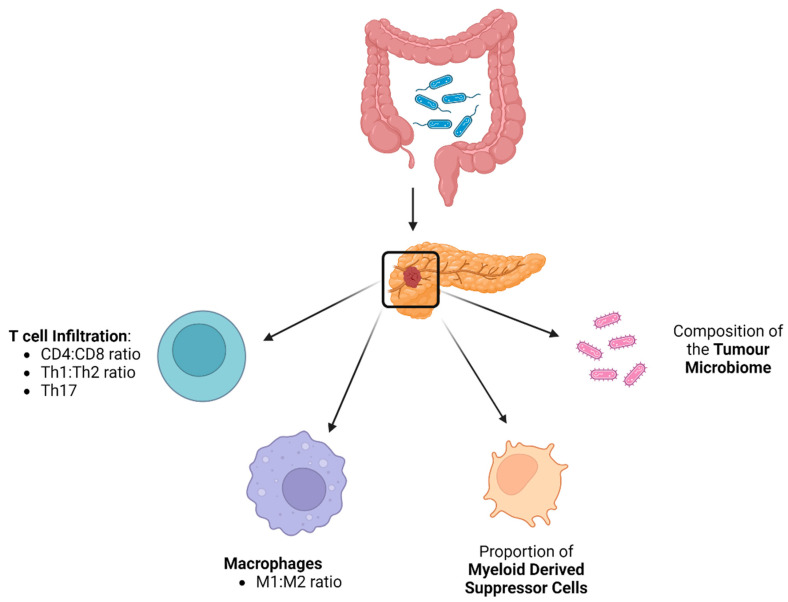
The gut microbiome influences the immune infiltration of PDAC tumours in multiple different ways (Created with BioRender.com).

**Figure 2 cancers-16-00996-f002:**
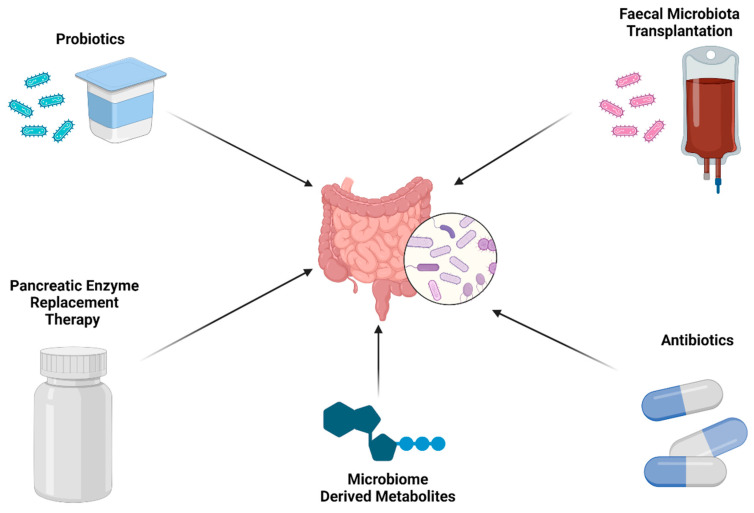
The microbiome could be modified towards a more favourable phenotype in PDAC (created with BioRender.com).

## Data Availability

The data can be shared up on request.

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
