# Peer review of "Involvement of the Gut Microbiome in the Local and Systemic Immune Response to Pancreatic Ductal Adenocarcinoma"

_cancers, 2024, doi:10.3390/cancers16050996_

Round 1

Reviewer 1 Report

Comments and Suggestions for Authors

Title is suggested to be changed to “Involvement of gut Microbiome in the Local and Systemic Immune Response in Pancreatic Ductal Adenocarcinoma”. 

Despite the subjects of this article sound interesting and clinically important, the following aspects shall be further addressed.

1.    Statement in Lines 29 - 32 “examines the mechanisms by which the gut microbiome has been shown to modulate the immune response to PDAC” is weakly proven.  More so, the argument lies between gut microbiome acts as a driver or a passenger of PDAC progression.  The latter seemed to be better supported by parts of manuscript.  For example, statement “PERT has been shown to reverse changes to the microbiome caused by PEI [107,108]“ (line 330) and another statement in lines 162-166.

2.    Published citations are predominantly in the phrases of “association or correlation” but minimally (if any) direct evidence denoting the impact of gut microbiomes on human diseases. Antibiotic administration seems to be insufficient to correlate the two and is not administration feasible as the long-term antibiotic treatment is unrealistic.  Solid evidence shall be evolved from sorting oncogenic apart from the tumor-suppressive microbiotas.

3.    As a general comment, while immune evasion is applicable to all cancers, PDAC is difficult to be treated is more due to dense desmoplastic stroma, early spread, and chemo treatment resistant, rather than aberrant gut microbiome.

4.    To align with text, Fig 1 shall be presented with hierarchy tiers between tumor gut microbioms (upper) and the other 3 immune suppressions (lower), rather than with a parallel fashion.

5.    Despite antibiotic ablation of gut microbiomes impact CD8 T lymphocytes, Th17, Th1/Th2 ratio etc, the oncogenic impact of gut microbiomes on PDAC remain unclear, because the former might exert a temporary effect rather than sustaining long-term impact on malignant progression.

6.    Molecular mechanisms linking microbiomes to T cell activation, macrophages differentiations, Th17 regulation, PRR expression, and Th1/Th2 ratio shall be delineated. 

7.    TAM influences immune cell infiltration through various immune cell differentiation. Line 248 shall be re-checked for correctness. A phrase shall be clarified as being “tumor-suppressive M1”.

8.    Quality of Figures 1 & 2 is only moderate.

The major pitfall of this paper is that, perhaps, gut microbiome is a consequence of rather than a driver of PDAC.

Comments on the Quality of English Language

English writing requires extensive improvement.

Author Response

Title is suggested to be changed to “Involvement of gut Microbiome in the Local and Systemic Immune Response in Pancreatic Ductal Adenocarcinoma”. 

  • Thank you for your suggestion, we have changed the title as you recommend.

Despite the subjects of this article sound interesting and clinically important, the following aspects shall be further addressed.

  1. Statement in Lines 29 - 32 “examines the mechanisms by which the gut microbiome has been shown to modulate the immune response to PDAC” is weakly proven.  More so, the argument lies between gut microbiome acts as a driver or a passenger of PDAC progression.  The latter seemed to be better supported by parts of manuscript.  For example, statement “PERT has been shown to reverse changes to the microbiome caused by PEI [107,108]“ (line 330) and another statement in lines 162-166.

- Thank you for your comment, we have softened the language in line 29 to account for the lack of certainty in this area and hope that this is satisfactory

- Regarding the second point, whether the gut microbiome is a driver of passenger of PDAC progression, this is an important question and we do not think that it is possible to yet answer this question given that evidence is still at an early stage. However the evidence presented in this review, which is solely looking at the relationship between the gut microbiome and the immune response to PDAC, indicates that the composition of the gut microbiome in animal models can influence the immune peripheral immune system and the immune infiltration of the tumour. Further human studies are required.

- Regarding PEI and PERT influencing the gut microbiome, not all PDAC patients will have PEI, but in those that do, it is possible that changes to the gut microbiome may contribute to the poorer prognosis amongst those with PDAC and untreated PEI. As such, further investigation into these associations in PDAC are required.

  1. Published citations are predominantly in the phrases of “association or correlation” but minimally (if any) direct evidence denoting the impact of gut microbiomes on human diseases. Antibiotic administration seems to be insufficient to correlate the two and is not administration feasible as the long-term antibiotic treatment is unrealistic.  Solid evidence shall be evolved from sorting oncogenic apart from the tumor-suppressive microbiotas.

- Thank you for your comment. Most of the studies on this subject are animal studies and in order to change the composition of the gut microbiome, the authors used antibiotics. We agree that this would not necessarily be clinically feasible but the aim of these studies was to demonstrate a relationship between the gut microbiome and the immune response to PDAC, rather than to identify clinically appropriate methods to alter the gut microbiome, This will come in later studies.  

  1. As a general comment, while immune evasion is applicable to all cancers, PDAC is difficult to be treated is more due to dense desmoplastic stroma, early spread, and chemo treatment resistant, rather than aberrant gut microbiome.

 - Thank you for your comment. We agree that there are many factors that make PDAC a difficult cancer to treat, so we have altered the language throughout the article to reflect this.

  1. To align with text, Fig 1 shall be presented with hierarchy tiers between tumor gut microbioms (upper) and the other 3 immune suppressions (lower), rather than with a parallel fashion.

- Thank you for your suggestion, the purpose of this figure was to summarise the four main mechanisms by which the gut microbiome influences  the immune infiltration of PDAC tumours. It has not been shown that the tumour microbiome is the mediator of these immune cell changes within the tumour so we are hesitant to rearrange the figure in that manner. We hope that is satisfactory and apologise if that was unclear.

  1. Despite antibiotic ablation of gut microbiomes impact CD8 T lymphocytes, Th17, Th1/Th2 ratio etc, the oncogenic impact of gut microbiomes on PDAC remain unclear, because the former might exert a temporary effect rather than sustaining long-term impact on malignant progression.

 - Thank you for your comment. We agree that the long-term impact is not clear and requires further investigation. Perhaps with the increasing availability of next generation sequencing, this will become more apparent over the next few years.

  1. Molecular mechanisms linking microbiomes to T cell activation, macrophages differentiations, Th17 regulation, PRR expression, and Th1/Th2 ratio shall be delineated. 

- Thank you for your comment. We agree that this is an important area to include in this review. In section 6.2, we discuss the pathways by which the bacteria of the microbiome are thought to influence the host immune system, for example through TLR9 and Dectin-1. We hope that is satisfactory.

  1. TAM influences immune cell infiltration through various immune cell differentiation. Line 248 shall be re-checked for correctness. A phrase shall be clarified as being “tumor-suppressive M1”.

- Thank you for your comment, we have adjusted line 248 as you suggest.

  1. Quality of Figures 1 & 2 is only moderate.

- Thank you for your comment, is this in reference to the figure itself or the resolution quality?

The major pitfall of this paper is that, perhaps, gut microbiome is a consequence of rather than a driver of PDAC.

Reviewer 2 Report

Comments and Suggestions for Authors

Interesting review.

However there are certain issues that we have to comment.

The following phrase:

Indeed, the diversity of the gut microbiome associates with short-term survivors (STS) and long-term survivors (LTS) in PDAC, with greater diversity linked to improved outcomes [21].

This comes from a report by Riquelme et al. and needs some more explanation. Reading your sentence is not understandable until you consult the original paper.

The following sentence is misleading:

There is a need for a greater understanding of the mechanisms behind the  profound immune evasion exhibited by PDAC tumours [24], which makes it one of the most challenging malignancies to treat .

You are meaning that PDAC is a challenge because of the immune evasion.

This is a flawed concept. PDAC is a challenge for many reasons including the immune problem. What about late diagnosis, lack of tumor markers, desmoplastic reaction, hypoxia, lack of drugs for KRAS mutation, difficulties in surgical access, high morbidity, drug resistance, etc.  

I do not understand the following sentence

The systemic immunosuppression exhibited by PDAC contributes significantly to its aggressive nature[40–42] and leads to its late presentation in most patients.

I do not understand what presents late: the aggressive nature, the immunosuppression, the diagnosis. Please clarify.

I do not agree with the following sentence

PDAC tumours progress rapidly due to immune evasion in the tumour microenvironment [40,52].

PDAC development requires between 15 to17 years to become a fully invasive tumor since the first driving mutation takes place. Therefore, there is no rapid progress. Furthermore, duplication time for PDAC cells is longer than for many other tumors such as lung adenocarcinoma, cervix, melanoma, breast cancer, etc. So  I would not use the word rapidly.

Immune evasion lets tumor develop, but progression depends from the genetic signature of the malignancy, not from immune evasion.

The following sentence is misleading unless you clarify:

Given that Th17 cells have been associated with neoplastic transformation  of pancreatic tumours [58]

The first step in the neoplastic transformation is Kras mutation. Kras uses a pathway as described by McAllister, your reference 58:

Using a murine model of pancreatic intraepithelial neoplasia (PanIN), we found that Kras(G12D) induces expression of functional IL-17 receptors on PanIN epithelial cells and also stimulates infiltration of the pancreatic stroma by IL-17-producing immune cells.

What you call Future Directions should be better called Discussion

The following article should be discussed in the paper. 

Tintelnot, J., Xu, Y., Lesker, T.R. et al. Microbiota-derived 3-IAA influences chemotherapy efficacy in pancreatic cancer. Nature 615, 168–174 (2023). https://doi.org/10.1038/s41586-023-05728-y

Another important omission is about bacteria that produce lactic acid n tumors such as Lactobacillus iners and increase resistance to chemo-radiotherapy and immune defenses.

Author Response

nteresting review.

However there are certain issues that we have to comment.

The following phrase:

Indeed, the diversity of the gut microbiome associates with short-term survivors (STS) and long-term survivors (LTS) in PDAC, with greater diversity linked to improved outcomes [21]. This comes from a report by Riquelme et al. and needs some more explanation. Reading your sentence is not understandable until you consult the original paper.

  • Sorry for this, we have rephrased this to make it clearer to the reader

The following sentence is misleading:

There is a need for a greater understanding of the mechanisms behind the  profound immune evasion exhibited by PDAC tumours [24], which makes it one of the most challenging malignancies to treat .

You are meaning that PDAC is a challenge because of the immune evasion.

This is a flawed concept. PDAC is a challenge for many reasons including the immune problem. What about late diagnosis, lack of tumor markers, desmoplastic reaction, hypoxia, lack of drugs for KRAS mutation, difficulties in surgical access, high morbidity, drug resistance, etc.  

  • Thank you for your comment. We can see how this can be misinterpreted so have altered this to make it clear that the immune invasion is one of many factors making PDAC difficult to treat

I do not understand the following sentence

The systemic immunosuppression exhibited by PDAC contributes significantly to its aggressive nature[40–42] and leads to its late presentation in most patients.

I do not understand what presents late: the aggressive nature, the immunosuppression, the diagnosis. Please clarify.

  • Thank you for your comment, to avoid confusion we have removed the part of the sentence referring to a ‘late presentation’

I do not agree with the following sentence

PDAC tumours progress rapidly due to immune evasion in the tumour microenvironment [40,52].

PDAC development requires between 15 to17 years to become a fully invasive tumor since the first driving mutation takes place. Therefore, there is no rapid progress. Furthermore, duplication time for PDAC cells is longer than for many other tumors such as lung adenocarcinoma, cervix, melanoma, breast cancer, etc. So  I would not use the word rapidly.

Immune evasion lets tumor develop, but progression depends from the genetic signature of the malignancy, not from immune evasion.

  • Thank you for your comment. We have altered this sentence to make it clearer and closer to what we were trying to convey: that the immune invasion in the tumour microenvironment is one of the contributing factors to poor prognosis in PDAC. We hope this is satisfactory

The following sentence is misleading unless you clarify:

Given that Th17 cells have been associated with neoplastic transformation  of pancreatic tumours [58]

The first step in the neoplastic transformation is Kras mutation. Kras uses a pathway as described by McAllister, your reference 58:

Using a murine model of pancreatic intraepithelial neoplasia (PanIN), we found that Kras(G12D) induces expression of functional IL-17 receptors on PanIN epithelial cells and also stimulates infiltration of the pancreatic stroma by IL-17-producing immune cells.

  • Thank you for pointing this out, we can see how this was not clear. We have adjusted the language here to more accurately reflect the findings of the study.

What you call Future Directions should be better called Discussion

  • Thank you, we have changed this as you suggest

The following article should be discussed in the paper. 

Tintelnot, J., Xu, Y., Lesker, T.R. et al. Microbiota-derived 3-IAA influences chemotherapy efficacy in pancreatic cancer. Nature 615, 168–174 (2023). https://doi.org/10.1038/s41586-023-05728-y

  • Thank you for highlighting this important study, We have added this along with other related studies into a new section, 7.2.

Another important omission is about bacteria that produce lactic acid n tumors such as Lactobacillus iners and increase resistance to chemo-radiotherapy and immune defenses.

  • Thank you for your comment. We have added a new paragraph to section 4 to discuss the effect of bacterial metabolites, such as lactic acid, on the immune response to PDAC

Reviewer 3 Report

Comments and Suggestions for Authors

The article “How Does the Gut Microbiome Influence the Local and Sys- 2 temic Immune Response to Pancreatic Ductal Adenocarcinoma?” provides a comprehensive review of the gut microbiome's role in modulating immune responses in the context of pancreatic ductal adenocarcinoma (PDAC). The authors explore the intricate relationship between gut microbiota and immune system development, emphasizing the influence of the gut microbiome on systemic immune response and PDAC tumor progression and response to treatment. The topic is timely and significant, considering the growing interest in the microbiome's role in cancer. The authors have done commendable work in consolidating a range of studies to highlight this complex interaction.

However,

-        it lacks a broader comparative analysis with other cancer types. It would be crucial to understand whether the phenomena the Authors describe are unique to PDAC or part of a broader pattern observable in other cancers (like neuroendocrine tumors-PMID: 35203339, or cholangiocarcinoma, or even prostate or colorectal cancers). Such a comparison is not just academically enriching but crucial for understanding the uniqueness or commonality of these mechanisms across different cancers.

-        The manuscript primarily references animal studies or in vitro research, it might lack a discussion on human studies or patient-based studies. This is important for establishing the clinical relevance of the findings. Please add a comment in the discussion.

-        There may be a lack of discussion regarding potential confounding factors that could influence the relationship between the gut microbiome and PDAC. This includes endoscopic procedures (such as ERCP), environmental factors, and the use of medications/antibiotics.

-        Moreover, even if the discussion on the impact of the gut microbiome on systemic immune response and PDAC tumor microenvironment is particularly insightful, the implications of these findings on current clinical practice are not thoroughly explained (see only as an example PMID: 36309208, PMID: 35440726 or PMID: 36646684). Please expand on the discussion regarding clinical implications and potential therapeutic strategies.

-        While the manuscript may suggest future areas of study, it might not provide a clear, actionable research pathway or identify specific gaps that need to be addressed in future work.

Comments on the Quality of English Language

The manuscript is generally well-written

Author Response

The article “How Does the Gut Microbiome Influence the Local and Sys- 2 temic Immune Response to Pancreatic Ductal Adenocarcinoma?” provides a comprehensive review of the gut microbiome's role in modulating immune responses in the context of pancreatic ductal adenocarcinoma (PDAC). The authors explore the intricate relationship between gut microbiota and immune system development, emphasizing the influence of the gut microbiome on systemic immune response and PDAC tumor progression and response to treatment. The topic is timely and significant, considering the growing interest in the microbiome's role in cancer. The authors have done commendable work in consolidating a range of studies to highlight this complex interaction.

Thank you for your kind comments

However,

-        it lacks a broader comparative analysis with other cancer types. It would be crucial to understand whether the phenomena the Authors describe are unique to PDAC or part of a broader pattern observable in other cancers (like neuroendocrine tumors-PMID: 35203339, or cholangiocarcinoma, or even prostate or colorectal cancers). Such a comparison is not just academically enriching but crucial for understanding the uniqueness or commonality of these mechanisms across different cancers.

- Thank you for your suggestion. We agree that this adds helpful context so have added this to section 5.

-        The manuscript primarily references animal studies or in vitro research, it might lack a discussion on human studies or patient-based studies. This is important for establishing the clinical relevance of the findings. Please add a comment in the discussion.

- Thank you for your suggestion, this is an important point and we have added some sentences to the start of the discussion

-        There may be a lack of discussion regarding potential confounding factors that could influence the relationship between the gut microbiome and PDAC. This includes endoscopic procedures (such as ERCP), environmental factors, and the use of medications/antibiotics.

- Thank you for this important point, we have added some sentences in the discussion which highlight this point and the importance of controlling for these factors in future studies of PDAC.

-        Moreover, even if the discussion on the impact of the gut microbiome on systemic immune response and PDAC tumor microenvironment is particularly insightful, the implications of these findings on current clinical practice are not thoroughly explained (see only as an example PMID: 36309208, PMID: 35440726 or PMID: 36646684). Please expand on the discussion regarding clinical implications and potential therapeutic strategies.

- Thank you for this important comment. We have signposted this more clearly in the discussion (line 336 onwards) so that this is clearer to the reader and hope that this is satisfactory.  

-        While the manuscript may suggest future areas of study, it might not provide a clear, actionable research pathway or identify specific gaps that need to be addressed in future work.

- Thank you for your comment, we have made the future areas for study more obvious by changing the language in the discussion to make it clearer for the reader (lines 329, 343, 356, 361, 381.) We have also added a new paragraph discussing possible avenues for future study with the gut microbiome and immune response to PDAC in human patients from line 381 onwards.

Comments on the Quality of English Language

The manuscript is generally well-written

Round 2

Reviewer 1 Report

Comments and Suggestions for Authors

The currently revised manuscript is now acceptable for publication.

Reviewer 3 Report

Comments and Suggestions for Authors The manuscript has been sufficiently improved Comments on the Quality of English Language

Good